# COVID-19 and the risk and trajectory of frailty in community- or institution-dwelling individuals: Protocol for systematic review and meta-analysis

**Saurabh P. Mehta**[1], **Julie M. Faieta**[2], **Maria Chang Swartz**[3], **Emily W. Blevins**[4], **Ahmed M. Negm**[5], **Vanina P. M. Dal Bello-Haas**[6] *

**1** Physical Therapy Program, East Tennessee State University, Johnson City, Tennessee, United States of America, **2** Department of Rehab Science & Technology, University of Pittsburgh, Pittsburgh, Pennsylvania, United States of America, **3** Department of Pediatrics-Research, UT MD Anderson Cancer Center, Houston, Texas, United States of America, **4** Quillen College of Medicine Library, East Tennessee State University, Mountain Home, Tennessee, United States of America, **5** Department of Surgery, Foothills Medical Centre, University of Calgary, Calgary, Alberta, Canada, **6** School of Rehabilitation Science, McMaster University, Hamilton, Ontario, Canada

* vdalbel@mcmaster.ca

**Data Availability Statement:** No datasets were generated or analysed during the current study. All

## Abstract

The purpose of this paper is to describe a protocol for a systematic review (SR) and meta-analysis examining the associations between an episode of COVID-19 and trajectory as well as the risk of frailty. The protocol for this SR has been registered in the PROSPERO database (CRD42023468297) and conforms to the guidelines proposed by the Preferred Reporting Items for Systematic Reviews (PRISMA). The search strategy will involve retrieving literature from six different databases and will be guided by keywords encompassing population (community-dwelling or institution-dwelling adults), exposure (episode of COVID-19), and outcome (frailty). The citations retrieved from the search process will be screened for their eligibility. The risk of bias for the articles identified to be eligible for the review will be examined using the Quality In Prognosis Studies (QUIPS) tool. The Metafor package in R will be used for quantitative data synthesis of the literature. Grading of Recommendations Assessment, Development and Evaluation (GRADE) will be used to assess the quality and certainty of the body of evidence. This systematic review will provide crucial information to determine whether an episode of COVID-19 is associated with an increased risk of frailty. The results of this review will have significant clinical implications in mitigating the risk of frailty after COVID-19.

## Introduction

The acute impacts of COVID-19 are now more elucidated in our 4th year post pandemic. The long-term impacts in the form of sustained symptoms, exacerbation of comorbidities, and increased risk of subsequent disease development are still emerging. Certain populations who

relevant data from this study will be made available upon study completion.

**Funding:** The author(s) received no specific funding for this work.

**Competing interests:** The authors have declared that no competing interests exist.

are more vulnerable to contracting COVID-19, or experiencing worse outcomes, are also more at risk of developing frailty or experiencing poorer frailty-related outcomes. These include older adults and individuals with chronic comorbid conditions [1]. While we are still without uniform consensus on the broader definition of frailty, it can be described as a "*biologic syndrome of decreased reserve and resistance to stressors, resulting from cumulative declines across multiple physiologic systems, and causing vulnerability to adverse outcomes*" [2]. Despite the lack of a uniform definition, any relationship between COVID-19 and frailty outcomes is important to assess considering the growing aging population and projected increases in the number of people with chronic conditions [3]. Naturally, the COVID-19 pandemic has increased attention toward frailty throughout clinical research. Sciacchitano et al [1] point out a dramatically increased number of publications that incorporate the term frailty in recent post-pandemic years. The interest in the relationship between COVID-19 and frailty, however, may be falling short in the sense that most research attends to only one direction of the bidirectional relationship between these two conditions.

The relationship between viral infection and frailty is often described concerning the influence that frailty has on health outcomes after viral infection. Pott et al [4] investigated combined cases of RSV and influenza and reported that frailty, among several other risk factors, was associated with a heightened risk of poor health outcomes [4]. In their systematic review of the literature, Yang et al [5] report that frailty has a significant positive association with exacerbated adverse health outcomes among those with COVID-19. What is lesser reported is the impact of viral infection on the incidence and severity of frailty in at-risk populations. This systematic review is designed to address this gap in our understanding of the relationship between viral infection and frailty. Due to the prevalence and persistence of COVID-19 as well as its long-term impacts, we are specifically addressing the research question, "Does an episode of COVID-19 affect the risk and trajectory of frailty in community-dwelling or institution-dwelling individuals?"

## Materials and methods

The protocol for this systematic review and meta-analysis has been registered on the International Prospective Register of Systematic Reviews (CRD42023468297). This protocol is consistent with the Preferred Reporting Items for Systematic Reviews and Meta-Analyses (PRISMA) [6]. To ensure a high-quality research design, this methodology consists of a comprehensive literature search, clearly defined eligibility criteria to select relevant studies, appraising the quality of studies using a standardized tool, extracting data showing associations of COVID-19 on the trajectory of frailty using pre-designed data form, and analyzing data quantitatively with meta-analysis. The quality of the studies included in the review will be appraised using the Quality In Prognosis Studies (QUIPS) tool [7,8].

### Literature search

Study investigators and a medical librarian collaboratively developed the search strategy to locate primary studies that examined associations between an episode of COVID-19 and the risk and trajectory of frailty in community-dwelling or institutionalized elderly. The search strategy consisted of two broader themes namely COVID-19 and frailty. The search string for COVID-19 as the subject heading was developed using the guidelines provided by the Canadian Agency for Drugs and Technologies in Health to locate COVID-19 literature on all major databases [9]. The search was limited to literature published after October 31, 2019, to avoid studies that included patients with other respiratory infections including previous variants of Coronavirus. This was rationalized considering the first known case of COVID-19 was

documented in December 2019 [10]. Six databases will be searched including PubMed, Embase, the *Cochrane* Database of Systematic Reviews, Cochrane Central Register of Controlled Trials (CENTRAL), SportDiscus, and Cumulated Index to Nursing and Allied Health Literature (CINAHL). No other limiters such as restricting the search to a particular age group or geographic location will be placed. The detailed search strategy is shown in S1 Appendix. Apart from the approaches identified above, the bibliography of the studies included in this review will be scanned to determine any potential article relevant to this review.

## Eligibility criteria and identification of studies for the review

The eligibility of the studies retrieved from the literature search will be reviewed using predefined inclusion and exclusion criteria shown in Table 1.

These criteria encompass categories of type of patient population recruited, exposure, outcomes, types of studies to be included, and timeline for the outcome to emerge after exposure. In addition, the studies published in languages other than English will be excluded from this review. All citations found from the literature search will be exported to the Covidence platform. Two reviewers will independently review all the citations to determine their eligibility, first by reviewing titles/abstracts and then by reviewing full-text copies of the articles. A third reviewer will arbitrate disagreements when the primary reviewers cannot reach a consensus on determining the eligibility of citations. Agreement between two raters in determining the eligibility of articles will be assessed using kappa statistic (κ), where κ values between 0.60–0.79

**Table 1. Eligibility criteria for the studies to be included in this review.**

| Category for consideration | Criteria |
|---|---|
| Participants/Study population | *Inclusion*: All the following criteria will need to be met for inclusion.<br>• Adults who lived in their dwellings or lived independently in settings such as retirement homes or supported living environments<br>• Adults who underwent assessment of frailty within 12 months after an episode of COVID-19<br>• Adults who were known to be non-frail before the episode of COVID-19<br>*Exclusion*: Participants who were known to be frail before COVID-19 |
| Determination of COVID-19 or exposure | *Inclusion*: A documented history of COVID-19 within the 12 months preceding the assessment of frailty; OR being deemed to have long COVID-19<br>*Exclusion*: Participants had respiratory infection/exposure other than COVID-19 |
| Determination of frailty or outcome | *Inclusion*: Frailty was measured using the tools known to be most used and highly cited in the literature [11]<br>*Exclusion*: Participants were deemed frail using frameworks not described in the literature or are uncommon [11] |
| Types of study designs | *Inclusion*: Randomized controlled trials, prospective or retrospective cohort studies<br>*Exclusion*: Systematic reviews, meta-analyses, or scoping reviews Conference abstracts, case reports, case studies, short communications, letters to the editors, protocol papers, position papers, protocol papers, white papers, editorials, feasibility studies |
| Linearity of exposure (COVID-19) versus outcome (frailty) OR time for follow-up | *Inclusion*: Studies should establish the occurrence of frailty within 12 months after the episode of COVID-19<br>*Exclusion*: Participants in the study were not assessed for frailty within 12 months after the episode of COVID-19 |

**Table 2. Details of attributes to be collected during data extraction.**

| Categories | Attributes |
|---|---|
| Details of the study | Author and year of publication<br>Study design<br>Study objectives<br>Geographic location and context of the study |
| Description of the study population | Inclusion/exclusion criteria<br>Number of participants<br>Demographic details—age, sex |
| Exposure assessment | The approach used for determining COVID-19 episode (e.g. self-report, medical records, diagnostic tests)<br>Whether the episode required hospitalization |
| Outcome assessment | List of outcomes including determination of frailty<br>Test/measurement method used to assess outcomes including frailty<br>Reliability/validity of outcome measures<br>Blinding of outcome assessor concerning history of COVID-19 |
| Follow-up | Follow-up time interval<br>Retention of participants<br>Approach for handling missing data |
| Covariates or confounders | List of covariates considered in the statistical model (e.g. demographic, health history, living situation, hospitalization for COVID-19) |
| Statistical analyses | Statistical model used for assessing association/risk between COVID-19 episode and frailty<br>Index for ascertaining effect size |
| Results | A list of variables, including COVID-19, was found to be associated with the development of frailty<br>Effect size (e.g. regression coefficient, odds ratio, risk ratio) and associated p-value for the variables found to be significant |

will suggest moderate agreement, values between 0.80–0.90 will suggest strong agreement and those >0.90 will suggest almost perfect agreement [12].

## Data extraction

The data from each study will be extracted for categories of participants, study design, exposure, outcomes assessed, statistical analyses, and results. Specifically, we will collect data for the design and objective of the study, description of the study population, exposure assessment, follow-up period, outcome assessed, covariates analyzed, statistical analyses conducted, and results. Table 2 outlines specific attributes for which the data will be extracted for each of these categories.

## Assessment of methodological quality and risk of bias of studies, and certainty of evidence

The risk of bias (RoB) and quality of methodology for the studies included in the will be examined using the QUIPS tool [7,8]. Six domains included in the QUIPS tool to assess the quality of studies are study participation (selection bias), attrition (attrition bias), prognostic factor assessment (measurement bias), outcome assessment (measurement bias), confounding (confounder bias), and statistical analysis and reporting (bias in statistical analysis). The overall RoB for each study will be determined based on RoB in each of the six domains. Studies will be deemed to have low RoB if all six domains have low RoB, or only one domain has moderate RoB. The overall RoB for a study will be high if one or more domains have high RoB or ≥3 domains have moderate RoB. All other studies will be considered to have moderate RoB [7,8].

**Table 3. Quality and certainty of evidence grades.**

| Grade | Description |
|---|---|
| High | Very confident the true effect lies close to that of the estimate of the effect. |
| Moderate | Moderately confident in the effect estimate. The true effect is likely to be close to the estimate of the effect. There is a possibility that it is substantially different. |
| Low | Limited confidence in the effect estimate. The true effect may be substantially different from the estimate of the effect. |
| Very Low | Little confidence in the effect estimate: The true effect is likely to be substantially different from the estimate of effect. |

[a]Adapted from: GRADE Handbook, Section 5. Quality of Evidence; https://gdt.gradepro.org/app/handbook/handbook.html.

We will use Grading of Recommendations Assessment, Development and Evaluation (GRADE) [13] to assess the quality and certainty of the body of evidence. The GRADE approach considers five factors (limitations in study design or execution (risk of bias); inconsistency of results; indirectness of evidence; imprecision; and publication bias), resulting in one of four grades along a continuum [14] (Table 3).

## Data analysis

Measures of effect (relative risk ratio [RR] or odds ratio [OR] for categorical outcomes and beta coefficients for continuous outcomes) for assessing the relationship between the prognostic variable (COVID-19) and the outcome (risk of frailty) will be used for pursuing data synthesis. Meta-analysis of data will be conducted if a pool of studies has sufficient homogeneity. Homogeneity will be examined by reviewing the similarities in clinical attributes such as the determination of exposure variable (i.e. method used for diagnosing COVID-19), duration between assessment of exposure and outcome, and the type of frailty assessed (e.g. physical frailty only versus multiple attributes of frailty). Statistical heterogeneity ($I^2$) will be considered significant for $I^2$ values of $>50\%$ [15]. An inverse variance random-effects model will be used for pooling the data [15]. Measures of effect such as RR, OR, or Beta coefficients will be combined to derive an overall estimate of the association between COVID-19 and the risk of frailty. Where necessary, Beta coefficients will be converted to OR using approaches discussed in the Cochrane Handbook of Systematic Reviews [15]. While we have not planned subgroup analysis a priori, subgroup analysis may be planned based on sex (females versus males) and age ($<65$ years of age versus $\geq65$ years of age) should there be sufficient data available for these subgroups. The metafor package in R will be used for conducting meta-analysis [16].

## Ethics and dissemination

Approval from an institutional review board is not required for this protocol or systematic review. The results of this systematic review and meta-analysis will be disseminated in a peer-reviewed journal.

## The status and timeline of the study

The review has been initiated and will be completed within the next 8-months for dissemination.

## Discussion

The long-term health [17–19] and economic [20,21] consequences following an episode of COVID-19 are well documented. Especially in older adults, the COVID-19 episode can have severe effects including higher odds of institutionalization as well as mortality [22,23]. Frailty is a clinical syndrome characterized by decreased reserve across multiple physiological systems [24]. The relationship between COVID-19 episodes and frailty has been explored in literature [5,25]. Using systematic review and meta-analysis design, this review will examine the prevalent literature to determine whether having an episode of COVID-19 is associated with the development of frailty within 12 months after the episode. The review will provide crucial information regarding the profile of people who have a greater likelihood of developing frailty after COVID-19, which will facilitate timely management to decrease the risk of frailty. The results of this study will be important for healthcare providers, patients, and their families of patients in understanding individual risk for developing frailty after COVID-19 and preparing healthcare as well as social support systems to manage frailty.

Undertaking systematic review and meta-analysis requires that the researchers follow a robust methodology to improve internal validity as well as generalization of results. This review will follow recommendations from the Cochrane Collaboration for conducting systematic reviews of prognostic studies [26]. These recommendations offer guidelines for each step of the review from locating the evidence using comprehensive search methods, to appraising the evidence using standardized appraisal tools and using high-quality statistical approaches to synthesize the data for understanding the phenomenon under the investigation. The literature search strategy shown in S1 Appendix was developed in consultation with a medical librarian and will locate relevant literature from six different databases. The QUIPS tool that will be used for assessing the quality of studies provides a detailed assessment of the methodological quality of the prognostic study [8]. This tool is one of the recommended instruments by the Cochrane Prognosis Methods Group [27]. The other strength of this review is that the author team is a group of multidisciplinary scholars who collectively have considerable expertise in conducting similar reviews in the past.

While we feel confident about our methodological approach, we foresee some challenges that we will encounter throughout the different phases of this review. One of the eligibility criteria for inclusion of studies in this review is to ensure that the study sample did not have frailty before the episode of COVID-19, since pre-existing diagnosis of frailty may render the assessment of associations between COVID-19 and frailty meaningless. It is anticipated that some studies may not have considered the prevalence of frailty before the COVID-19 episode in the study sample, which would misclassify people and increase false positive cases in that study. To our end, we will locate and separately analyze studies where the sample was recruited from institutionalized settings such as long-term care or skilled nursing facilities since it is logical to assume that the prevalence of frailty before COVID-19 episode might be higher in such individuals. Secondly, we anticipate significant diversity in the timing and measurement tools used for determining frailty. Especially, a wide range of assessment tools has been identified in the literature [11] due to the broad interpretation of what frailty is [24]. Our approach to only include studies that have used the most commonly utilized tool will lead to some clarity and increased generalizability of this review. Lastly, multiple indices are used to demonstrate associations between two variables [28]. There is a possibility that the use of different indices across different studies included in this review will render meta-analysis very challenging. We, however, do hope that the binary nature of both the exposure (i.e. episode of COVID-19 or not) and the outcome variable (occurrence of frailty or not) will bring some level of homogeneity in the statistical index used across the studies to demonstrate an association.

## Conclusion

While COVID-19 infections may have peaked worldwide, the disabling aftereffects of the infections continue to prevail in many individuals. Understanding the risk profile for developing adverse health outcomes such as frailty after a COVID-19 episode can improve care for COVID-19 survivors. The results of this systematic review will have important clinical implications for understanding the profile of patients at higher risk of developing frailty after an episode of COVID-19. Moreover, the study results will also identify prevalent gaps in knowledge to understand better the risk of developing frailty after an episode of COVID-19.

## Supporting information

**S1 Checklist. PRISMA-P (Preferred Reporting Items for Systematic review and Meta-Analysis Protocols) 2015 checklist: Recommended items to address in a systematic review protocol\*.**
(DOC)

**S1 Appendix.**
(DOCX)

## Author Contributions

**Conceptualization:** Saurabh P. Mehta, Julie M. Faieta, Maria Chang Swartz, Ahmed M. Negm, Vanina P. M. Dal Bello-Haas.

**Data curation:** Saurabh P. Mehta, Emily W. Blevins.

**Investigation:** Saurabh P. Mehta, Julie M. Faieta, Maria Chang Swartz, Emily W. Blevins, Ahmed M. Negm, Vanina P. M. Dal Bello-Haas.

**Methodology:** Saurabh P. Mehta, Julie M. Faieta, Maria Chang Swartz, Ahmed M. Negm, Vanina P. M. Dal Bello-Haas.

**Project administration:** Saurabh P. Mehta.

**Visualization:** Saurabh P. Mehta, Julie M. Faieta, Maria Chang Swartz, Ahmed M. Negm, Vanina P. M. Dal Bello-Haas.

**Writing – original draft:** Saurabh P. Mehta, Vanina P. M. Dal Bello-Haas.

**Writing – review & editing:** Saurabh P. Mehta, Julie M. Faieta, Maria Chang Swartz, Emily W. Blevins, Ahmed M. Negm, Vanina P. M. Dal Bello-Haas.

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
