## [Decision Letter · Decision Letter 0]

19 Jun 2024

PONE-D-24-12625Does an episode of COVID-19 affect the prevalence and trajectory of frailty in community- or institution-dwelling individuals? A protocol for systematic review and meta-analysisPLOS ONE

Dear Dr. Dal Bello-Haas,

Thank you for submitting your manuscript to PLOS ONE. After careful consideration, we feel that it has merit but does not fully meet PLOS ONE’s publication criteria as it currently stands. Therefore, we invite you to submit a revised version of the manuscript that addresses the points raised during the review process.

We look forward to receiving your revised manuscript.

Kind regards,

Sílvia Fernanda da Rocha-Rodrigues Mendes

Academic Editor

PLOS ONE

Journal Requirements:

Reviewers' comments:

Reviewer's Responses to Questions

**Comments to the Author**

1. Does the manuscript provide a valid rationale for the proposed study, with clearly identified and justified research questions?

Reviewer #1: Yes

2. Is the protocol technically sound and planned in a manner that will lead to a meaningful outcome and allow testing the stated hypotheses?

Reviewer #1: Yes

3. Is the methodology feasible and described in sufficient detail to allow the work to be replicable?

Reviewer #1: Yes

4. Have the authors described where all data underlying the findings will be made available when the study is complete?

Reviewer #1: Yes

5. Is the manuscript presented in an intelligible fashion and written in standard English?

Reviewer #1: Yes

6. Review Comments to the Author

You may also provide optional suggestions and comments to authors that they might find helpful in planning their study.

Reviewer #1: The authors present a systematic review protocol that aims to answer whether "an episode of COVID-19 affects the prevalence and trajectory of frailty in community-dwelling or institution-dwelling individuals".

The protocol was well designed, following PRISMA recommendations, but I suggest small revisions to make it better for publication.

The title as a question looks strange and I suggest that the authors evaluate the change to: COVID-19 and risk of frailty in

community- or institution-dwelling individuals: protocol for systematic review and meta-analysis. Changing the title, in lines 89 and 90 you should change " examined associations between an episode of COVID-19 and the prevalence and trajectory of frailty in community-dwelling or institutionalized elderly. " to examined associations between an episode of COVID-19 and the risk of frailty in community-dwelling or institutionalized elderly.

Still in the methods, the authors must change this criterion "In addition, the studies published in languages other than English will be excluded from this review", as with online translation tools and artificial intelligence, it is now possible to translate manuscripts written in languages other than English, preventing an important bias in the future systematic review.

Finally, they must include in the protocol that the certainty of the evidence for the main outcome, risk of frailty, will be assessed by GRADE.

7. PLOS authors have the option to publish the peer review history of their article (what does this mean?). If published, this will include your full peer review and any attached files.

Reviewer #1: **Yes: **Ricardo Ney Cobucci

---

## [Author Response · Author response to Decision Letter 0]

30 Jul 2024

RESPONSE TO REVIEWERS LETTER HAS BEEN UPLOADED - BELOW IS A CUT AND PASTE OF THE UPLOADED DOCUMENT:

July 20, 2024

Sylvia Fernanda da Rocha-Rodrriguez Mendes, Ph.D.

Academic Editor

PLOS ONE

Dear Dr. da Rocha-Rodrriguez Mendes,

On behalf of the co-authors, I would like to thank you and the Reviewer for the very thoughtful comments and feedback to enhance our paper. 

All comments received via email sent June 19, 2024 have been cut and pasted below in this letter. Our responses follow each comment received and are bolded and highlighted in yellow. 

Revisions within the manuscript are evident via track changes; and/or manuscript sections where the revisions were made are noted in this letter because page and line numbers have changed from the original submission, because of additions made, and page and line numbers differ between the marked-up and clean copies.

We believe the comments and feedback received and subsequent revisions have greatly improved our paper and we look forward to hearing from you soon.

Sincerely,

Vanina Dal Bello-Haas, PT, PhD

FEEDBACK RECEIVED

A. Journal Requirements:

1. Please ensure that your manuscript meets PLOS ONE's style requirements, including those for file naming. The PLOS ONE style templates can be found at  https://journals.plos.org/plosone/s/file?id=wjVg/PLOSOne_formatting_sample_main_body.pdf and  https://journals.plos.org/plosone/s/file?id=ba62/PLOSOne_formatting_sample_title_authors_affiliations.pdf

We have reviewed the information found at the links and believe we have adequately addressed PLOS ONE’s style requirements. 

Our paper is a systematic review protocol paper, there are no data. We noted this in our original submission; see Manuscript, pg 14.

3. Please review your reference list to ensure that it is complete and correct. If you have cited papers that have been retracted, please include the rationale for doing so in the manuscript text, or remove these references and replace them with relevant current references. Any changes to the reference list should be mentioned in the rebuttal letter that accompanies your revised manuscript. If you need to cite a retracted article, indicate the article’s retracted status in the References list and also include a citation and full reference for the retraction notice. 

We have reviewed and believe there are no issues. We have not cited any retracted papers. Note: we have added two references to address Reviewer’s comment regarding needing to add GRADE. 

B. Reviewers' comments:  Reviewer's Responses to Questions

Comments to the Author  

1. Does the manuscript provide a valid rationale for the proposed study, with clearly identified and justified research questions? The research question outlined is expected to address a valid academic problem or topic and contribute to the base of knowledge in the field.

Reviewer #1: Yes

2. Is the protocol technically sound and planned in a manner that will lead to a meaningful outcome and allow testing the stated hypotheses? The manuscript should describe the methods in sufficient detail to prevent undisclosed flexibility in the experimental procedure or analysis pipeline, including sufficient outcome-neutral conditions (e.g. necessary controls, absence of floor or ceiling effects) to test the proposed hypotheses and a statistical power analysis where applicable. As there may be aspects of the methodology and analysis which can only be refined once the work is undertaken, authors should outline potential assumptions and explicitly describe what aspects of the proposed analyses, if any, are exploratory.

Reviewer #1: Yes

3. Is the methodology feasible and described in sufficient detail to allow the work to be replicable? Descriptions of methods and materials in the protocol should be reported in sufficient detail for another researcher to reproduce all experiments and analyses. The protocol should describe the appropriate controls, sample size calculations, and replication needed to ensure that the data are robust and reproducible.

Reviewer #1: Yes

4. Have the authors described where all data underlying the findings will be made available when the study is complete? The PLOS Data policy requires authors to make all data underlying the findings described in their manuscript fully available without restriction, with rare exception, at the time of publication. The data should be provided as part of the manuscript or its supporting information, or deposited to a public repository. For example, in addition to summary statistics, the data points behind means, medians and variance measures should be available. If there are restrictions on publicly sharing data—e.g. participant privacy or use of data from a third party—those must be specified.

Reviewer #1: Yes

5. Is the manuscript presented in an intelligible fashion and written in standard English? PLOS ONE does not copyedit accepted manuscripts, so the language in submitted articles must be clear, correct, and unambiguous. Any typographical or grammatical errors should be corrected at revision, so please note any specific errors here.

Reviewer #1: Yes

Regarding #1 to #5 above - thank you.

6. Review Comments to the Author Please use the space provided to explain your answers to the questions above and, if applicable, provide comments about issues authors must address before this protocol can be accepted for publication. You may also include additional comments for the author, including concerns about research or publication ethics. You may also provide optional suggestions and comments to authors that they might find helpful in planning their study.   (Please upload your review as an attachment if it exceeds 20,000 characters)

Reviewer #1: The authors present a systematic review protocol that aims to answer whether "an episode of COVID-19 affects the prevalence and trajectory of frailty in community-dwelling or institution-dwelling individuals". The protocol was well designed, following PRISMA recommendations, but I suggest small revisions to make it better for publication. The title as a question looks strange and I suggest that the authors evaluate the change to: COVID-19 and risk of frailty in community- or institution-dwelling individuals: protocol for systematic review and meta-analysis. 

Thank you for the positive comment. 

We have revised the title of the manuscript as suggested by changing to risk; and, the title is no longer in the form of a question. Our preference is to keep trajectory [course over time] in the title as this is also our focus/what we are interested in examining. 

Changing the title, in lines 89 and 90 you should change " examined associations between an episode of COVID-19 and the prevalence and trajectory of frailty in community-dwelling or institutionalized elderly. " to examined associations between an episode of COVID-19 and the risk of frailty in community-dwelling or institutionalized elderly. 

We have made this change; also, please see above. Note: we have also changed prevalence to risk in other sections of the manuscript as appropriate. 

Still in the methods, the authors must change this criterion "In addition, the studies published in languages other than English will be excluded from this review", as with online translation tools and artificial intelligence, it is now possible to translate manuscripts written in languages other than English, preventing an important bias in the future systematic review. 

Thank you for this suggestion. 

We agree including English only research articles will be a limitation. We will address this accordingly in our final manuscript.

The recommendation regarding using AI/online translation tools is interesting. We are hesitant to make this change and have provided our rationales for not doing so below:

1. The protocol was registered [International Prospective Register of Systematic Reviews (CRD42023468297)]. Our protocol underwent review - only COVID-related protocols underwent peer review at the time of us submitting our registration documents. The reviewers did not make any comments regarding English only articles nor made any recommendations to use AI for translation. 

2. We are aware of only a few papers that have used AI for conducting literature searches e.g., Alaniz et al. (2023; The Utility of Artificial Intelligence for Systematic Reviews and Boolean Query Formulation and Translation). We are not aware of any systematic reviews that have used online tools/AI for translation purposes for systematic reviews. Thus, there is no precedence or related processes that have been published/are available that would serve as a guide. 

3. As noted by Alaniz et al. (2023; above) many LLM’s training datasets have a very recent cut-off date (e.g., 2021) – this would seriously constrain conducting a comprehensive search, also resulting in incomplete information for our systematic review. 

4. Younis et al’s 2024 paper [A Systematic Review and Meta-Analysis of Artificial Intelligence Tools in Medicine and Healthcare: Applications, Considerations, Limitations, Motivation and Challenges] highlights that papers published to date describe theoretical potentials of AI tools rather than practical implementations and effective strategies to address real-world challenges, such as correct interpretations, accuracy and reliability. 

5. According to El Hawary et al (2023) researchers should always verify AI-assisted work and references to ensure both accuracy and relevance. Similarly, work undertaken with online translation tools may have errors, particularly for more complex material or specialized subject matter e.g., 57.7% accuracy for medical phrase translation (Patil & Davies 2014); translation accuracy rates ranged from 55% to 94% depending on the language (Taira et al 2021). Using these tools would also require verification to ensure correctness and precision of the translation. 

Our research team does not have the human or financial resources that would be required to ensure relevance, accuracy and precision of AI-assisted/on-line tool translation of articles published in languages other than English.

Finally, they must include in the protocol that the certainty of the evidence for the main outcome, risk of frailty, will be assessed by GRADE.

This addition has been made Note: we also added this to the abstract to ensure consistency between the abstract and the manuscript.

7. PLOS authors have the option to publish the peer review history of their article (what does this mean?). If published, this will include your full peer review and any attached files. Do you want your identity to be public for this peer review? For information about this choice, including consent withdrawal, please see our Privacy Policy.

Reviewer #1: Yes: Ricardo Ney Cobucci

Thank you Dr. Cobucci.

---

## [Decision Letter · Decision Letter 1]

2 Oct 2024

COVID-19 and the risk and trajectory of frailty in community- or institution-dwelling individuals: Protocol for systematic review and meta-analysis.

PONE-D-24-12625R1

Dear Dr.  PM Dal Bello-Haas,

We’re pleased to inform you that your manuscript has been judged scientifically suitable for publication and will be formally accepted for publication once it meets all outstanding technical requirements.

Kind regards,

Milad Khorasani, PhD

Academic Editor

PLOS ONE

Additional Editor Comments (optional):

Reviewers' comments:

Reviewer's Responses to Questions

**Comments to the Author**

1. Does the manuscript provide a valid rationale for the proposed study, with clearly identified and justified research questions?

Reviewer #1: Yes

2. Is the protocol technically sound and planned in a manner that will lead to a meaningful outcome and allow testing the stated hypotheses?

Reviewer #1: Yes

3. Is the methodology feasible and described in sufficient detail to allow the work to be replicable?

Reviewer #1: Yes

4. Have the authors described where all data underlying the findings will be made available when the study is complete?

Reviewer #1: Yes

5. Is the manuscript presented in an intelligible fashion and written in standard English?

Reviewer #1: Yes

6. Review Comments to the Author

You may also provide optional suggestions and comments to authors that they might find helpful in planning their study.

Reviewer #1: The authors have met most of the recommendations and the manuscript is of improved quality. Congratulations.

7. PLOS authors have the option to publish the peer review history of their article (what does this mean?). If published, this will include your full peer review and any attached files.

Reviewer #1: **Yes: **Ricardo Ney Cobucci

---

## [Editor Report · Acceptance letter]

6 Nov 2024

PONE-D-24-12625R1 

PLOS ONE

Dear Dr. Dal Bello-Haas, 

I'm pleased to inform you that your manuscript has been deemed suitable for publication in PLOS ONE. Congratulations! Your manuscript is now being handed over to our production team.

Kind regards, 

on behalf of

Dr. Milad Khorasani 

Academic Editor

PLOS ONE